

Comparing observed and modelled components of the Atlantic Meridional Overturning
Circulation at 26°N.
Harry Bryden[1], Sybren Drijfhout[1,2], Jennifer Mecking[3], Wilco Hazeleger[2]
[1]Ocean and Earth Science, University of Southampton, Southampton United Kingdom
[2]Faculty of Geosciences, University of Utrecht, Utrecht, The Netherlands
[3]National Oceanography Centre, Southampton, United Kingdom
Correspondence Email: hlb@soton.ac.uk
15 November 2023
**Abstract**
The Coupled Model Intercomparison Project (CMIP) allows assessment of the representation
of the Atlantic Meridional Overturning Circulation (AMOC) in climate models. While CMIP
Phase 6 models display a large spread in AMOC strength by a factor of three, the multi-model
mean strength agrees reasonably well with observed estimates from RAPID[1], but this does not
hold for its various components. In CMIP6 the present-day AMOC is characterised by a lack
of lower North Atlantic Deep Water (lNADW), due to the small-scale of Greenland-Iceland-
Scotland Ridge overflow and too much mixing. This is compensated by increased
recirculation in the subtropical gyre and more Antarctic Bottom Water (AABW). Deep-water
circulation is dominated by a distinct deep western boundary current (DWBC) with minor
interior recirculation compared to observations. The future decline in the AMOC to 2100 of
7Sv under a SSP5-8.5 scenario is associated with decreased northward western boundary
current transport in combination with reduced southward flow of upper North Atlantic Deep
Water (uNADW). In CMIP6, wind stress curl decreases with time by 14% so that the wind-
driven thermocline recirculation in the subtropical gyre is reduced by 4 Sv (17%) by 2100.
The reduction in western boundary current transport of 11Sv is more than the decrease in the
wind-driven gyre transport suggesting a decrease over time in the component of the Gulf
Stream originating in the South Atlantic.
**1. Introduction**
The Atlantic Meridional Overturning Circulation (AMOC) is the Atlantic part of the global
overturning circulation. Our understanding of the strength, variability and structure of the
AMOC has improved since the deployment of the RAPID[1] array, which monitors the volume
transport at 26°N since April 2004 (Moat et al., 2020). Additionally, these observations serve
as invaluable reference data for the representation of the AMOC in coupled climate and Earth
System models. The most recent phase of the Coupled Model Intercomparison Project, CMIP
Phase 6, allows us to assess the representation of the AMOC in these models. The models
project the AMOC strength will decline over the next century (Lee et al., 2021). Here we
compare observed and modeled components of the AMOC over the historical period 2004 to
2014 and then assess how the ensemble-mean CMIP6 transport components change in a
declining AMOC over the next century under SSP5-8.5 emission scenario.

---

[1] RAPID is used here as shorthand for the RAPID-Meridional Overturning Circulation and Heatflux
Array-Western Boundary Time Series at 26°N (Moat et al., 2022).



The RAPID AMOC observations from 2004 to 2018 indicate that the AMOC has declined by
2.4 Sv, about 12%, from 18.3 Sv to 15.9 Sv (Bryden, 2021).  The decline is primarily evident
in reduced southward transport of lower North Atlantic Deep Water (lNADW) that is
balanced by slightly reduced Gulf Stream transport and more southward recirculation within
the subtropical gyre.  In CMIP6 models, the AMOC declines by about 40% over the 21st
century (Weijer et al., 2020).  Here we analyse 19 CMIP6 model projections in order to
identify which components lead to the AMOC decline, for clues as to how the AMOC may
change within the continuing RAPID observational framework.
The Coupled Model Intercomparison Project (CMIP) is a comprehensive effort of modelling
centres around the world to improve our understanding about past, present and future changes
of the climate system (Eyring et al., 2016; O'Neill et al., 2016). Even though CMIP6 shows
improvements compared to previous CMIP generations, model biases related to the AMOC
persist. These include a shallow bias to the deep cell, too much deep convection, and a too-
small temperature difference between its upper and lower limbs.  Additionally, CMIP6
models largely underestimate low-frequency variability of the AMOC and show large inter-
model differences in their AMOC representation (Weijer et al., 2020).
The RAPID array monitors the AMOC volume transport at 26°N since April 2004 (Smeed et
al., 2018). The transport through the cross section is estimated by a decomposition of the
AMOC into 3 components: (1) transport through the Florida Straits, (2) Ekman surface
transport generated by zonal wind stress, and (3) density driven interior transport estimated
from mooring measurements.  The mid-ocean interior transport is further broken down into
thermocline recirculation (0-800m depth)), intermediate water transport (800-1100m), upper
North Atlantic Deep Water (1100-3000m), lower North Atlantic Deep Water (3000-5000m).
The goal of this study is to gain insight into the cause of disagreement between CMIP6
models and RAPID data in terms of AMOC strength, structure and variability.  We
decompose the modelled AMOC transport at 26°N from CMIP6 into the same transport
components as measured by the RAPID array.  We compare the CMIP6 transport components
with the observed Rapid components for the historical period 2004-2014.  We then examine
the change of these components in CMIP6 under the SSP5-8.5 emission scenario from the
historical period until 2100.
**2.  Data and Methods**
Monthly averages of AMOC estimates from the RAPID array are compared to the historical
simulations of 19 CMIP6 models.  Note that only the overlapping period was investigated,
April 2004 – December 2014.  Details of the 19 CMIP6 models are given in Table 1.  The
SSP5-8.5 future projection from 2015 to 2100, is then used to investigate how the AMOC
may change in future projections.  For each model, one ensemble member was used.



Table 1: Metadata and references of the models analysed in this study. References are from the Earth System Grid Federation

| Model | Modelling centre | Horizontal resolution (°) | Variant label | Data reference historical | Data reference SSP585 |
|---|---|---|---|---|---|
| CAMS-CSM1-0 | CAMS | 1 x 1 | r1i1p1f1 | Rong (2019) | Rong (2019) |
| CAS-ESM2-0 | CAS | 1 x 1 | r1i1p1f1 | Chai (2020) | Unknown (2018) |
| CESM2-WACCM | NCAR | 1 x 1 | r1i1p1f1 | Danabasoglu (2019) | Danabasoglu (2019) |
| CIESM | THU | 1 x 1 | r1i1p1f1 | Huang (2019) | Huang (2020) |
| CMCC-CM2-SR5 | CMCC | 1 x 1 | r1i1p1f1 | Lovato and Peano (2020) | Lovato and Peano (2020) |
| CMCC-ESM2 | CMCC | 1 x 1 | r1i1p1f1 | Lovato et al. (2021) | Lovato et al. (2021) |
| CNRM-CM6-1 | CNRM-CERFACS | 1 x 1 | r1i1p1f2 | Voldoire (2019) | Voldoire (2019) |
| CNRM-ESM2-1 | CNRM-CERFACS | 1 x 1 | r2i1p1f2 | Seferian (2018) | - |
| CanESM5 | CCCma | 1 x 1 | r1i1p1f1 | Swart et al. (2019) | Swart et al. (2019) |
| EC-Earth3 | EC-Earth Consortium | 1 x 1 | r1i1p1f1 | EC-Earth Consortium (2021) | EC-Earth Consortium (2019) |
| FIO-ESM-2-0 | FIO-QLNM | 1 x 1 | r1i1p1f1 | Song et al. (2019) | Song et al. (2019) |
| HadGEM3-GC31-LL | MOHC | 1 x 1 | r1i1p1f3 | Ridley et al. (2019) | Good (2020) |
| HadGEM3-GC31-MM | MOHC | 0.25 x 0.25 | r1i1p1f3 | Ridley et al. (2019) | Ridley et al. (2019) |
| IPSL-CM6A-LR | IPSL | 1 x 1 | r1i1p1f1 | Boucher et al. (2021) | Boucher et al. (2019) |
| MPI-ESM1-2-HR | MPI | 0.4 x 0.4 | r1i1p1f1 | Jungclaus et al. (2019) | Schupfner et al. (2019) |
| MPI-ESM1-2-LR | MPI | 1.5 x 1.5 | r1i1p1f1 | Wieners et al. (2019) | Wieners et al. (2019) |
| MRI-ESM2-0 | MRI | 1 x 0.5 | r1i1p1f1 | Yukimoto et al. (2019) | Yukimoto et al. (2019) |
| NESM3 | NUIST | 1 x 1 | r1i1p1f1 | Cao and Wang (2019) | Cao (2019) |
| UKESM1-0-LL | MOHC | 1 x 1 | r1i1p1f2 | Tang et al. (2019) | Good et al. (2019) |

A cross section between Florida and the African continent at the latitude closest to 26°N was
selected for each model. The net transport through the section, approximately -1 Sv for each
model due to the Bering Strait throughflow, was removed before computing the AMOC
components from meridional velocities as follows:
Florida Straits Transport (FS): CMIP6 models do not resolve the Bahama Islands and as a
result the Florida Straits proper. For this reason the following definition is used. The
boundary between Florida Straits (FS) transport and mid-ocean transport is defined as the
longitude where the depth-averaged transport (from the surface down to the depth of
maximum overturning) changes from positive (northward) to negative (southward). This
definition thus identifies the FS transport as the western boundary current, thereby including
the transport by the Antilles Current, which in CMIP6 models cannot be separated from the
Florida Current.
Thermocline Recirculation (tr): East of FS and from the surface to down to the depth of
horizontally averaged potential temperature of 8°C.
Intermediate Waters (iw): East of FS and between the depth of horizontally averaged potential
temperature of 8°C and depth of maximum overturning.
Upper North Atlantic Deep Water (uNADW): Between the depth of maximum overturning
and the depth of horizontally averaged potential temperature of 3°C.
Lower North Atlantic Deep Water (lNADW): Between the depth of horizontally averaged
potential temperature of 3°C and the depth where horizontally-averaged transport changes
from negative to positive.
Antarctic Bottom Water (AABW): Between the depth where horizontally-averaged transport
changes from negative to positive and the bottom.




Ekman (ek): Near surface ageostrophic transport estimated from the zonal wind stress.

Multi-model means (MMM) for each component over the 19 models are then made with their
standard deviation.

**3. Results**

Figure 1 compares the RAPID observations of the AMOC transport components with the
CMIP6 components for the historical period 2004-2014. For the historical period (2004-
2014) the MMM CMIP6 AMOC underestimates the observed AMOC transport by 2.2 Sv
(Table 2). The underestimation of AMOC strength in the CMIP6 models is likely related to
the reduced transport of lower NADW, due to the small scale of Greenland-Iceland-Scotland
Ridge overflow compared to the resolution of models and excessive mixing at this location. In
a study of deep waters in CMIP6, Heuzé (2021) noted that the models did form water masses
similar in properties to lNADW in the Nordic Seas, but none of the deep waters made it over
the ridge and into the Iceland or Irminger basins. In the models, this lack of lNADW is
partially compensated by increased southward flow of upper NADW so the total southward
flow of deep water in CMIP6 is comparable to that observed by RAPID. The variability of
NADW is underestimated, most likely due to the inability of models to reproduce lower
NADW overflow. Deep-water circulation in models is dominated by a distinct DWBC with
minor interior recirculation compared with observations. CMIP6 MMM Florida Straits (FS)
transport (37.4 Sv) is larger than observed Florida Straits transport (31.3 Sv). The relatively
coarse-resolution models do not resolve the narrow Florida Straits, and the model western
boundary current includes the narrow Antilles Current east of the Bahamas as well as the Gulf
Stream flow through Florida Straits. Recent estimates of Antilles Current transport are about
5 Sv (Meinen et al., 2019) and adding this transport to the observed Florida Straits transport
suggests that the observed (36.3 Sv) and modeled (37.4 Sv) western boundary current
transports are similar. The low-frequency variability of Florida Straits transport is largely
overestimated in CMIP6 models and we hypothesize that the inclusion of the Antilles Current
in this component in models may be a significant contributor to this variability as the
observed Antilles Current transport exhibits rms variability of 10 Sv that is not correlated
with Florida Straits transport variability. The MMM thermocline recirculation (tr) in CMIP6
models (-26.2 Sv) is larger than observed by the RAPID array (-18.6 Sv) though again this
may be due to issues on how the Antilles Current transport is accounted in the observations
and in the models. RAPID estimates thermocline recirculation to be the overall southward
flow between the Bahamas and Africa and this overall flow includes both the Antilles Current
transport and the mid-ocean thermocline recirculation associated with the wind-driven
subtropical gyre. If we separate out the northward Antilles Current transport of 5 Sv, then the
mid-ocean thermocline recirculation for RAPID would be -23.6 Sv (Table 2) in more
reasonable agreement with the CMIP6 MMM thermocline circulation of -26.2 Sv. Overall,
the MMM circulation in CMIP6 models for the historical period reasonably represents the
observed circulation in RAPID except for the underestimated lNADW transport associated
with issues of model representation of flows over ridges.



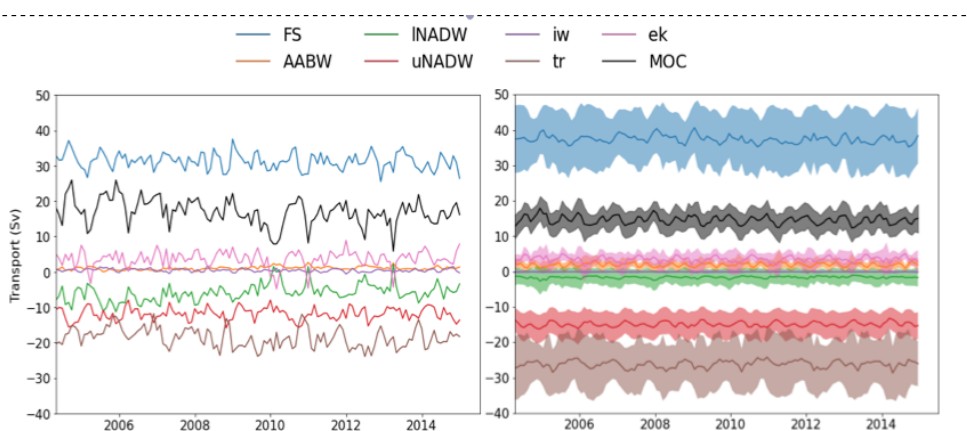

Figure 1. Historical time series for RAPID data (left) and multi-model mean CMIP6 data (right). Shaded areas indicate one standard deviation of the ensemble spread. This is Figure 6 in Beunk (2022)

Table 2. Components of the Atlantic Meridional Overturning Circulation at 26°N

|  | Rapid (2004-14) | CMIP6 Average | | |
|  |  | Historical (2004-14) | 2090-2100 | Decline |
| --- | --- | --- | --- | --- |
| **Upper Water** |  |  |  |  |
| Florida Straits (FS) | 31.3 |  |  |  |
| Ekman | 3.6 | 3.5 | 3.4 | 0.1 (1%) |
| Intermediate Water (IW) | 0.4 | ---- | ---- |  |
| Thermocline Recirculation (TR) | -18.6 |  |  |  |
| AMOC=FS+Ekman+IW+TR | 16.7 |  |  |  |
|  |  |  |  |  |
| Antilles Current (AC) | 5 |  |  |  |
| Western Boundary Current (FS+AC) | 36.3 | 37.4 | 26.4 | 11 (30%) |
| Thermocline Recirculation +AC | -23.6 |  |  |  |
| Model Thermocline Recirculation |  | -26.2 | -21.8 | 4.4 (17%) |
| Western Boundary Current+Ekman+Model TR | 14.7 | 8.0 |  | 6.7 (45%) |
|  |  |  |  |  |
| **Deep Water** |  |  |  |  |
| uNADW | -11.9 | -14.9 | -9.9 | 5.0 (34%) |
| lNADW | -5.9 | -1.6 | -0.2 | 1.4 (85%) |
| AABW | 1.0 | 1.9 | 2.1 | -0.2 (9%) |
| AMOC=uNADW+lNADW+AABW | -16.8 | -14.6 | -8.0 | 6.6 (45%) |

CMIP6 model projections suggest that the AMOC will decline over the next century as noted by Weijer et al. (2020). Here we find that the AMOC declines by 45% over the period 2015 to 2100 in a MMM of 19 CMIP6 projections. For comparison, over the RAPID time period 2004 to 2021, the AMOC has exhibited a small (order 12%) reduction that is manifest principally in reduced southward transport of lNADW (Bryden, 2021). It is of interest to identify which components contribute to the projected 45% decline in the AMOC over the coming century in CMIP6 simulations.

All 19 CMIP6 models analysed here exhibit a decline in the AMOC over the 21st century



(Table 3). This decline of the AMOC under SSP5-8.5 is in line with other modelling studies
(Levang and Schmitt, 2020; Weijer et al, 2020; Roberts et al., 2020). Averaged over the 19
models, the AMOC decline from 2004-2014 to 2090-2100 is 6.6 Sv or 45% in the AMOC
transport for the historical period (Figure 2). We find that the decline in the AMOC at 26°N
in CMIP6 models from 2015 to 2100 is dominated by a 30% decrease in western boundary
current transport (FS in Figure 2) and a 34% reduction in southward deep water transport
(uNADW in Figure 2). As Ekman transport (ek) shows no significant change in the model
projections, the AMOC decline of 6.6 Sv in the upper waters is the result of the difference
between the decline in western boundary current (FS) transport of 11.0 Sv and the 17%
decline in southward thermocline recirculation (tr) of 4.4 Sv. For the lower waters the overall
decline in northward transport of upper waters of 6.6 Sv is compensated by a decrease in
uNADW transport of 6.4 Sv and a small increase in northward AABW transport of 0.2 Sv, so
that the net transport through the cross section remains zero.

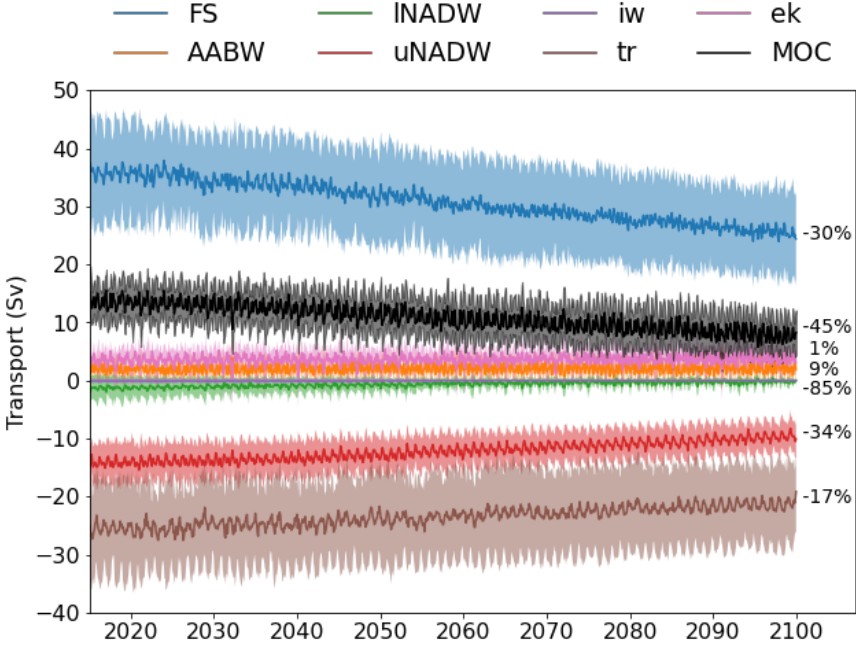

Figure 2. Multi-model mean timeseries of each component under SSP5-8.55. Shaded areas
illustrate one standard deviation of the inter-model spread. Percentages show the decline
relative to the historical period. This is Figure 12 In Beunk (2022).



| Model name | Historical mean (Sv) | 2090-2100 mean (Sv) | Change (Sv) | Change (%) |
|---|---|---|---|---|
| CAMS-CSM1-0 | 12.4 | 8.9 | -3.5 | -28 |
| CAS-ESM2-0 | 18.4 | 13.7 | -4.7 | -26 |
| CESM2-WACCM | 17.9 | 6.8 | -11.1 | -62 |
| CIESM | 11.4 | 4 | -7.4 | -65 |
| CMCC-CM2-SR5 | 14.2 | 9.2 | -5.0 | -35 |
| CMCC-ESM2 | 13.3 | 9.3 | -4.0 | -30 |
| CNRM-CM6-1 | 15.7 | 6.9 | -8.8 | -56 |
| CNRM-ESM2-1 | 15.3 | | | |
| CanESM5 | 11.4 | 5.5 | -5.9 | -52 |
| EC-Earth3 | 16.2 | 10.7 | -5.5 | -34 |
| FIO-ESM-2-0 | 17.7 | 10.7 | -7.0 | -39 |
| HadGEM3-GC31-LL | 15.2 | 7.9 | -7.3 | -48 |
| HadGEM3-GC31-MM | 15.4 | 6.5 | -8.9 | -58 |
| IPSL-CM6A-LR | 11.6 | 6.5 | -5.1 | -44 |
| MPI-ESM1-2-HR | 14.8 | 8.6 | -6.2 | -42 |
| MPI-ESM1-2-LR | 16.6 | 11.4 | -5.2 | -31 |
| MRI-ESM2-0 | 15.4 | 5 | -10.4 | -67 |
| NESM3 | 9.0 | 5 | -4.0 | -45 |
| UKESM1-0-LL | 15.6 | 7.8 | -7.8 | -50 |

Table 3. Values of the total AMOC for every model. Shown are the historical mean values,
2090-2100 mean values, absolute change and relative change. Changes are relative to the
historrical period. This Table is Appendix G in Beunk (2022).
To examine changes in wind-driven circulation over the 21$^{st}$ century in the subtropical North
Atlantic, we examined the mean wind-stress curl along the 26°N section for the historical and
SSP585 period. The values are negative (i.e. clockwise rotation), which results in southward
mid-ocean Sverdrup transport. Since the upper level gyre circulation is driven by wind-stress
curl (DiNezio et al., 2009; Zhao and Johns, 2014), we expect a decrease of this driver to affect
both Florida Straits transport and thermocline recirculation. Averaged over the model
projections, wind stress curl decreases by 14% from about 6 x 10$^{-8}$m s$^{-2}$. On the basis of
Sverdrup dynamics, we expect this change in wind stress curl will reduce the thermocline
recirculation at 26°N and indeed the thermocline recirculation does decrease by 4.4 Sv or
17% over the 21$^{st}$ century. We conclude that the reduction of thermocline recirculation is
almost entirely caused by a decline in wind-stress curl. On the basis of western
intensification theory (Stommel, 1948), the decrease in wind-stress curl should also lead to a
decrease in western boundary current transport by a similar amount. Thus we can explain a
decrease in western boundary current transport of 4.4 Sv over the 21st century as being due to
changes in the wind forcing.
The change in the western boundary current transport of 11 Sv in the CMIP projections is due
to a reduction in the wind-driven component by 4.4 Sv and to a reduction in the component of
the Gulf Stream flow originating from the South Atlantic of 6.6 Sv. The overall 6.6 Sv
reduction in the northward flow in the upper waters is then compensated by a reduction in
southward flow of the deep waters. In CMIP6, the reduction in the southward flow of deep
water is almost entirely due to a decreased DWBC transport of uNADW over the period
2015-2100. Thus the projected AMOC reduction over the 21st century in CMIP6 is due to a
reduction in the thermohaline circulation where there is less northward transport of upper
waters principally in the western boundary current across 26°N and less southward deep water
transport in the deep western boundary current.



## 4. Discussion

Over the SSP5-8.5 period (2015-2100) in CMIP6 projections, we find declines in the western boundary current transport, thermocline recirculation and NADW transport. Decreased thermocline recirculation is related to a decline in wind stress curl along the section and this decline is also expected to contribute to the decline in Gulf Stream transport. But the decline in western boundary current transport in CMIP6 models is substantially greater than the decline in wind stress curl and accompanying thermocline recirculation. Therefore, for the upper water circulation the CMIP6 decline in the AMOC is mostly caused by a decrease in the component of the western boundary current associated with the thermohaline circulation. For the lower water circulation, the decline in southward transport over the SSP5-8.5 period is associated with reduced uNADW transport. The overall reduction in southward deep water transport suggests a decline in NADW formation.

In a similar study, Asbjornsen and Arthun (2023) examined future changes in the AMOC using 14 CMIP6 models and found a weakening AMOC by 8.5 Sv over the coming century. For their ensemble, the Gulf Stream weakened by 33% or 11.2 Sv, 3.7 Sv of which was due to change in wind stress, and the Deep Western Boundary Current transport weakened by 8.5 Sv. As noted above, the CMIP6 projections are consistent in projecting a decline in the AMOC this century (Table 3), but the exact size of the AMOC reduction depends on which models are used for the study.

Because the AMOC is responsible for most of the northward heat transport in the Atlantic Ocean (Johns et al., 2011; Johns et al., 2023), CMIP6 model projections also exhibit a decrease in northward heat transport at 26°N over the 2015-2100 time period (Mecking and Drijfhout, 2023). The northward ocean heat transport across 26°N decreases by an average of 0.3 PW for the SSP5-8.5 scenario and this represents a 30% decline from the historical value of 1.0 PW.

The decline in the thermohaline circulation at 26°N implies that the overturning circulation south of 26°N, that is in the global circulation outside the North Atlantic, has also changed. The extra-Atlantic circulation converts deep water into upper and intermediate waters so that the southward deep water flow across 26°N and out of the North Atlantic must ultimately be converted within the global ocean into upper and intermediate waters that flow back into the North Atlantic and northward across 26°N. The decline in the North Atlantic thermohaline circulation at 26°N suggests that this global-scale overturning circulation must also have changed. Baker et al (2023) have explored how 2 mechanisms converting deep water into upper water south of 26°N change within CMIP6 simulations. The 2 mechanisms considered are Southern Ocean upwelling associated with eastward wind stress around Antarctica (Toggweiler and Samuels, 1993) and Indo-Pacific diffusive upwelling associated with deep interior mixing (Munk, 1966). Baker et al. found that the wind stress around Antarctica did not decline enough to account for a reduced 6 Sv upwelling of deep water, in fact there appeared to be a small increase in Southern Ocean wind stress and upwelling. Instead they found evidence in the CMIP6 projections that the interior Indo-Pacific upwelling declined enough to account for reduced conversion of deep waters into thermocline waters. They attributed such decline to the global warming that increases stratification (Li et al., 2020) and inhibits vertical mixing and associated upwelling.

Overall, the Atlantic and global overturning circulations appear to have declined in CMIP6 projections from 2015 to 2100. The manifestation of these declines at 26°N include a



reduction in the southward transport of NADW and a compensating reduction in the
northward flow of upper and thermocline waters through Florida Straits. The reduction in
southward deep water transport in CMIP6 is linked to a lack of lNADW formed in the Nordic
Seas flowing out over the Greenland-Iceland-Scotland Ridge into the northern Atlantic
(Heuzé, 2021); and the reduction in northward flow of upper waters is linked to a decrease in
diffusive upwelling in the Indo-Pacific related to increased stratification due to global
warming (Li et al., 2020; Baker et al., 2023). The ability of coupled climate models to
realistically include these critical processes of deep water formation, mixing in ridge
overflows and mid-ocean diffusive upwelling for future projections of ocean circulation
should be carefully assessed. In particular, the representation of deep water formation
in coupled climate models could be examined in comparison with observed production of
deep water. Implementing mixing parameterisations for overflows (Holt et al., 2017) in
coupled climate models could be assessed for their effectiveness in allowing the southward
transport of lNADW into and through the North Atlantic. And coupled climate models could
be examined for their parameterisations of diffusive mixing and upwelling, testing how
different parameterisations affect the global ocean overturning circulation over century time
scales.

In terms of observations, our results suggest that the ongoing RAPID project should
separately measure the Antilles Current and add it to Florida Straits transport for a true
measure of western boundary current transport for comparison with modelled transport
components. And the Antilles Current transport should be separated from the net mid-ocean
southward flow across 26N in the upper 800m that RAPID labels thermocline recirculation so
as to identify the actual mid-ocean thermocline recirculation associated with the wind stress
curl. By separately estimating the Antilles Current transport contribution, the RAPID project
could then provide well-defined estimates for the wind-driven and thermohaline contributions
to the AMOC at 26°N.

**Code Availability**

The code used to obtain the results of this study and a file containing metadata of the models
is freely available on GitHub: https://github.com/jordibeunk/MSc_Thesis.git

**Data Availability**
RAPID data and notes are freely available at
https://rapid.ac.uk/rapidmoc/rapid_data/datadl.php
19 CMIP6 models are used. The choice of these models is motivated by the fact that both
historical (2004-20015) data and future (2015-2100) projections under Shared Socioeconomic
Pathway 5-8.5 are available for all used variables. The model data has been accessed through
the Centre for Environmental Data Analysis (CEDA) archive https://data.ceda.ac.uk
**Author Contributions**

This work is based on an MSc thesis by Jordi Beunk at Utrecht University. Jennifer Mecking,
Sybren Drijfhout and Harry Bryden designed the project. Sybren Drijfhout and Wilco
Hazeleger identified the student and supervised the project in Utrecht while Mecking and
Bryden provided advice during the project and write-up of the thesis. After finishing the
thesis, Jordi Beunk indicated that he did not wish to be involved in writing up the results for



publication. Harry Bryden prepared a draft for this paper based on Beunk's thesis. Drijfhout,
Mecking and Hazeleger then edited the draft and all authors added elements of discussion
related to recent papers based on CMIP6 results.
**Competing interests**
The contact author declares that none of the authors has any competing interests
**Acknowledgments**
Bryden was a lead investigator for the NERC-funded project that first deployed the
transocean Rapid instrument array in 2004 under grant NER/T/S/2002/00481 and he has
continued to carry out analyses involving the ongoing Rapid observations following formal
retirement in 2011. Drijfhout and Mecking have been funded by NERC under the Wider
Impacts of Subpolar North Atlantic decadal variability on the ocean and atmosphere
(WISHBONE) grant NE/T0133478/1.

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

decades of Atlantic Ocean mass and heat transports at 26.5 N. *Philosophical Transactions of the
Royal Society A*, **381**(2262).
Jungclaus, Johann; Bittner, Matthias; Wieners, Karl-Hermann; Wachsmann, Fabian; Schupfner,
Martin; Legutke, Stephanie; Giorgetta, Marco; Reick, Christian; Gayler, Veronika; Haak, Helmuth; de
Vrese, Philipp; Raddatz, Thomas; Esch, Monika; Mauritsen, Thorsten; von Storch, Jin-Song; Behrens,
Jörg; Brovkin, Victor; Claussen, Martin; Crueger, Traute; Fast, Irina; Fiedler, Stephanie; Hagemann,
Stefan; Hohenegger, Cathy; Jahns, Thomas; Kloster, Silvia; Kinne, Stefan; Lasslop, Gitta; Kornblueh,
Luis; Marotzke, Jochem; Matei, Daniela; Meraner, Katharina; Mikolajewicz, Uwe; Modali,
Kameswarrao; Müller, Wolfgang; Nabel, Julia; Notz, Dirk; Peters-von Gehlen, Karsten; Pincus, Robert;
Pohlmann, Holger; Pongratz, Julia; Rast, Sebastian; Schmidt, Hauke; Schnur, Reiner; Schulzweida,
Uwe; Six, Katharina; Stevens, Bjorn; Voigt, Aiko; Roeckner, Erich (2019). MPI-M MPI-ESM1.2-HR
model output prepared for CMIP6 CMIP historical. Version 20211003.Earth System Grid Federation.
https://doi.org/10.22033/ESGF/CMIP6.6594
Lee, J.-Y., J. Marotzke, G. Bala, L. Cao, S. Corti, J.P. Dunne, F. Engelbrecht, E. Fischer, J.C. Fyfe, C.
Jones, A. Maycock, J. Mutemi, O. Ndiaye, S. Panickal, and T. Zhou, 2021: Future Global Climate:
Scenario-Based Projections and Near- Term Information. In Climate Change 2021: The Physical
Science Basis. Contribution of Working Group I to the Sixth Assessment Report of the
Intergovernmental Panel on Climate Change [Masson-Delmotte, V., P. Zhai, A. Pirani, S.L. Connors, C.
Péan, S. Berger, N. Caud, Y. Chen, L. Goldfarb, M.I. Gomis, M. Huang, K. Leitzell, E. Lonnoy, J.B.R.
Matthews, T.K. Maycock, T. Waterfield, O. Yelekçi, R. Yu, and B. Zhou (eds.)]. Cambridge University
Press, Cambridge, United Kingdom and New York, NY, USA, pp. 553–672,
doi:10.1017/9781009157896.006.
Levang, S. J., & Schmitt, R. W. (2020). What Causes the AMOC to Weaken in CMIP5?. *Journal of
Climate*, *33*(4), 1535-1545.
Li, G., L. Cheng, J.Zhu, K. E. Trenberth, M. E. Mann and J. P. Abraham (2020) Increasing ocean
stratification over the past half-century. *Nat. Clim. Change* **10**, 1116–1123.
Lovato, Tomas; Peano, Daniele (2020). CMCC CMCC-CM2-SR5 model output prepared for CMIP6
CMIP historical. Version 20211003.Earth System Grid Federation.
https://doi.org/10.22033/ESGF/CMIP6.3825
Lovato, Tomas; Peano, Daniele (2020). CMCC CMCC-CM2-SR5 model output prepared for CMIP6
ScenarioMIP ssp585. Version 20211003.Earth System Grid Federation.
https://doi.org/10.22033/ESGF/CMIP6.3896
Lovato, Tomas; Peano, Daniele; Butenschön, Momme (2021). CMCC CMCC-ESM2 model output
prepared for CMIP6 CMIP historical. Version 20211003.Earth System Grid Federation.



https://doi.org/10.22033/ESGF/CMIP6.13195

Lovato, Tomas; Peano, Daniele; Butenschön, Momme (2021). CMCC CMCC-ESM2 model output
prepared for CMIP6 ScenarioMIP ssp585. Version 20211003.Earth System Grid Federation.
https://doi.org/10.22033/ESGF/CMIP6.13259

McCarthy, G. D., Smeed, D. A., Johns, W. E., Frajka-Williams, E., Moat, B. I., Rayner, D., ... & Bryden,
H. L. (2015). Measuring the Atlantic meridional overturning circulation at 26 N. *Progress in*
*Oceanography*, *130*, 91-111.

Mecking, J.V. and Drijfhout, S. S. (2023).  The decrease in ocean heat transport in response to global
warming.  *Nature Climate Change*, pp. 1-8.

Meinen, C. S., Johns, W. E., Moat, B. I., Smith, R. H., Johns, E. M., Rayner, D., ... & Garzoli, S. L. (2019).
Structure and variability of the Antilles Current at 26.5 N. *Journal of Geophysical Research:*
*Oceans*, *124*(6), 3700-3723.

Moat B. I. et al. (2020) Pending recovery in the strength of the meridional overturning circulation at
26°N. *Ocean Sci*. 16, 863–874. (doi:10.5194/os-16-863-2020).

Moat, B. I., Frajka-Williams, E., Smeed, D. A., Rayner, D., Johns, W. E., Baringer, M. O., et al. (2022).
Atlantic meridional overturning circulation observed by the RAPID-MOCHA-WBTS (RAPID-meridional
Overturning Circulation and Heatflux Array-Western Boundary Time Series) array at 26N from 2004
to 2020 (v2020.2). [Dataset]. British Oceanographic Data Centre, Natural Environment Research
Council. https://doi.org/10.5285/e91b10af-6f0a-7fa7-e053-6c86abc05a09

Munk, W., 1966: Abyssal recipes. *Deep-Sea Res.*, **13,** 707-730.

O'Neill, B. C., Tebaldi, C., Vuuren, D. P. V., Eyring, V., Friedlingstein, P., Hurtt, G., et al. (2016). The
scenario model intercomparison project (ScenarioMIP) for CMIP6. Geoscientific Model Development,
9(9), 3461–3482. https://doi.org/10.5194/gmd-9-3461-2016

Ridley, Jeff; Menary, Matthew; Kuhlbrodt, Till; Andrews, Martin; Andrews, Tim (2019). MOHC
HadGEM3-GC31-LL model output prepared for CMIP6 CMIP historical. Version 20211003.Earth
System Grid Federation. https://doi.org/10.22033/ESGF/CMIP6.6109

Ridley, Jeff; Menary, Matthew; Kuhlbrodt, Till; Andrews, Martin; Andrews, Tim (2019). MOHC
HadGEM3-GC31-MM model output prepared for CMIP6 CMIP historical. Version 20211003.Earth
System Grid Federation. https://doi.org/10.22033/ESGF/CMIP6.6112

Roberts, M. J., Jackson, L. C., Roberts, C. D., Meccia, V., Docquier, D., Koenigk, T., ... & Wu, L. (2020).
Sensitivity of the Atlantic meridional overturning circulation to model resolution in CMIP6
HighResMIP simulations and implications for future changes. *Journal of Advances in Modeling Earth*
*Systems*, *12*(8), e2019MS002014.

Rong, Xinyao (2019). CAMS CAMS_CSM1.0 model output prepared for CMIP6 CMIP historical.
Version 20211003. Earth System Grid Federation. https://doi.org/10.22033/ESGF/CMIP6.9754

Rong, Xinyao (2019). CAMS CAMS-CSM1.0 model output prepared for CMIP6 ScenarioMIP ssp585.
Version 20211003.Earth System Grid Federation. https://doi.org/10.22033/ESGF/CMIP6.11052

Schupfner, Martin; Wieners, Karl-Hermann; Wachsmann, Fabian; Steger, Christian; Bittner, Matthias;





Jungclaus, Johann; Früh, Barbara; Pankatz, Klaus; Giorgetta, Marco; Reick, Christian; Legutke,
Stephanie; Esch, Monika; Gayler, Veronika; Haak, Helmuth; de Vrese, Philipp; Raddatz, Thomas;
Mauritsen, Thorsten; von Storch, Jin-Song; Behrens, Jörg; Brovkin, Victor; Claussen, Martin; Crueger,
Traute; Fast, Irina; Fiedler, Stephanie; Hagemann, Stefan; Hohenegger, Cathy; Jahns, Thomas;
Kloster,Silvia; Kinne, Stefan; Lasslop, Gitta; Kornblueh, Luis; Marotzke, Jochem; Matei, Daniela;
Meraner,Katharina; Mikolajewicz, Uwe; Modali, Kameswarrao; Müller, Wolfgang; Nabel, Julia; Notz,
Dirk; Peters-von Gehlen, Karsten; Pincus, Robert; Pohlmann, Holger; Pongratz, Julia; Rast, Sebastian;
Schmidt, Hauke; Schnur, Reiner; Schulzweida, Uwe; Six, Katharina; Stevens, Bjorn; Voigt, Aiko;
Roeckner, Erich (2019). DKRZ MPI-ESM1.2-HR model output prepared for CMIP6 ScenarioMIP
ssp585. Version 20211003.Earth System Grid Federation.
https://doi.org/10.22033/ESGF/CMIP6.4403
Seferian, Roland (2018). CNRM-CERFACS CNRM-ESM2-1 model output prepared for CMIP6 CMIP
historical. Version 20211003.Earth System Grid Federation.
https://doi.org/10.22033/ESGF/CMIP6.4068
Smeed, D.A., S. A. Josey, C. Beaulieu, W.E. Johns, B. I. Moat, E. Frajka-Williams, D. Rayner, C. S.
Meinen, M. O. Baringer, H. L. Bryden, and G. D. McCarthy. 2018. The North Atlantic Ocean is in a
state of reduced overturning, *Geophysical Research Letters*, 45,
https://doi.org/10.1002/2017GL076350.
Song, Zhenya; Qiao, Fangli; Bao, Ying; Shu, Qi; Song, Yajuan; Yang, Xiaodan (2019). FIO-QLNM FIO-
ESM2.0 model output prepared for CMIP6 CMIP historical. Version 20211003.Earth System Grid
Federation. https://doi.org/10.22033/ESGF/CMIP6.9199
Song, Zhenya; Qiao, Fangli; Bao, Ying; Shu, Qi; Song, Yajuan; Yang, Xiaodan (2019). FIO-QLNM FIO-
ESM2.0 model output prepared for CMIP6 ScenarioMIP ssp585. Version 20211003.Earth System Grid
Federation. https://doi.org/10.22033/ESGF/CMIP6.9214
Stommel, H. (1948). The westward intensification of wind-driven ocean currents. *Eos, Transactions
American Geophysical Union*, *29*(2), 202-206.
Swart, Neil Cameron; Cole, Jason N.S.; Kharin, Viatcheslav V.; Lazare, Mike; Scinocca, John F.; Gillett,
Nathan P.; Anstey, James; Arora, Vivek; Christian, James R.; Jiao, Yanjun; Lee, Warren G.; Majaess,
Fouad; Saenko, Oleg A.; Seiler, Christian; Seinen, Clint; Shao, Andrew; Solheim, Larry; von Salzen,
Knut; Yang, Duo; Winter, Barbara; Sigmond, Michael (2019). CCCma CanESM5 model output
prepared for CMIP6 CMIP historical. Version 20211003.Earth System Grid Federation.
https://doi.org/10.22033/ESGF/CMIP6.3610
Swart, Neil Cameron; Cole, Jason N.S.; Kharin, Viatcheslav V.; Lazare, Mike; Scinocca, John F.; Gillett,
Nathan P.; Anstey, James; Arora, Vivek; Christian, James R.; Jiao, Yanjun; Lee, Warren G.; Majaess,
Fouad; Saenko, Oleg A.; Seiler, Christian; Seinen, Clint; Shao, Andrew; Solheim, Larry; von Salzen,
Knut; Yang, Duo; Winter, Barbara; Sigmond, Michael (2019). CCCma CanESM5 model output
prepared for CMIP6 ScenarioMIP ssp585. Version 20211003.Earth System Grid Federation.
https://doi.org/10.22033/ESGF/CMIP6.3696
Tang, Yongming; Rumbold, Steve; Ellis, Rich; Kelley, Douglas; Mulcahy, Jane; Sellar, Alistair; Walton,
Jeremy; Jones, Colin (2019). MOHC UKESM1.0-LL model output prepared for CMIP6 CMIP
historical. Version 20211003.Earth System Grid Federation.
https://doi.org/10.22033/ESGF/CMIP6.6113



Toggweiler, J. R., & Samuels, B. (1998). On the ocean's large-scale circulation near the limit of no
vertical mixing. *Journal of Physical Ocean- ography*, *28*(9), 1832–1852. https://doi.org/10.1175/1520-
579  0485(1998)028

Voldoire, Aurore (2019). CNRM-CERFACS CNRM-CM6-1-HR model output prepared for CMIP6 CMIP
historical. Version 20211003.Earth System Grid Federation.
https://doi.org/10.22033/ESGF/CMIP6.4067
Voldoire, Aurore (2019). CNRM-CERFACS CNRM-CM6-1-HR model output prepared for CMIP6
ScenarioMIP ssp585. Version 20211003.Earth System Grid Federation.
https://doi.org/10.22033/ESGF/CMIP6.4225
Weijer, W., Cheng, W., Garuba, O. A., Hu, A., & Nadiga, B. T. (2020). CMIP6 models predict significant
21st century decline of the Atlantic Meridional Overturning Circulation. *Geophysical Research*
*Letters*, *47*(12), e2019GL086075.
Wieners, Karl-Hermann; Giorgetta, Marco; Jungclaus, Johann; Reick, Christian; Esch, Monika; Bittner,
Matthias; Legutke, Stephanie; Schupfner, Martin; Wachsmann, Fabian; Gayler, Veronika; Haak,
Helmuth; de Vrese, Philipp; Raddatz, Thomas; Mauritsen, Thorsten; von Storch, Jin-Song; Behrens,
Jörg; Brovkin, Victor; Claussen, Martin; Crueger, Traute; Fast, Irina; Fiedler, Stephanie; Hagemann,
Stefan; Hohenegger, Cathy; Jahns, Thomas; Kloster, Silvia; Kinne, Stefan; Lasslop, Gitta; Kornblueh,
Luis; Marotzke, Jochem; Matei, Daniela; Meraner, Katharina; Mikolajewicz, Uwe; Modali,
Kameswarrao; Müller, Wolfgang; Nabel, Julia; Notz, Dirk; Peters-von Gehlen, Karsten; Pincus, Robert;
Pohlmann, Holger; Pongratz, Julia; Rast, Sebastian; Schmidt, Hauke; Schnur, Reiner; Schulzweida,
Uwe; Six, Katharina; Stevens, Bjorn; Voigt, Aiko; Roeckner, Erich (2019). MPI-M MPI-ESM1.2-LR
model output prepared for CMIP6 CMIP historical. Version 20211003.Earth System Grid
Federation. https://doi.org/10.22033/ESGF/CMIP6.6595
Wieners, Karl-Hermann; Giorgetta, Marco; Jungclaus, Johann; Reick, Christian; Esch, Monika; Bittner,
Matthias; Gayler, Veronika; Haak, Helmuth; de Vrese, Philipp; Raddatz, Thomas; Mauritsen,
Thorsten; von Storch, Jin-Song; Behrens, Jörg; Brovkin, Victor; Claussen, Martin; Crueger, Traute;
Fast, Irina; Fiedler, Stephanie; Hagemann, Stefan; Hohenegger, Cathy; Jahns, Thomas; Kloster, Silvia;
Kinne, Stefan; Lasslop, Gitta; Kornblueh, Luis; Marotzke, Jochem; Matei, Daniela; Meraner, Katharina;
Mikolajewicz, Uwe; Modali, Kameswarrao; Müller, Wolfgang; Nabel, Julia; Notz, Dirk; Peters-von
Gehlen, Karsten; Pincus, Robert; Pohlmann, Holger; Pongratz, Julia; Rast, Sebastian; Schmidt, Hauke;
Schnur, Reiner; Schulzweida, Uwe; Six, Katharina; Stevens, Bjorn; Voigt, Aiko; Roeckner,
Erich (2019). MPI-M MPI-ESM1.2-LR model output prepared for CMIP6 ScenarioMIP
ssp585. Version 20211003.Earth System Grid Federation.
https://doi.org/10.22033/ESGF/CMIP6.6705
Yan, X., Zhang, R., & Knutson, T. R. (2018). Underestimated AMOC variability and implications for
AMV and predictability in CMIP models. *Geophysical Research Letters*, *45*(9), 4319-4328.
Yukimoto, Seiji; Koshiro, Tsuyoshi; Kawai, Hideaki; Oshima, Naga; Yoshida, Kohei; Urakawa, Shogo;
Tsujino, Hiroyuki; Deushi, Makoto; Tanaka, Taichu; Hosaka, Masahiro; Yoshimura, Hiromasa; Shindo,
Eiki; Mizuta, Ryo; Ishii, Masayoshi; Obata, Atsushi; Adachi, Yukimasa (2019). MRI MRI-ESM2.0 model
output prepared for CMIP6 CMIP historical. Version 20211003.Earth System Grid
Federation. https://doi.org/10.22033/ESGF/CMIP6.6842
Yukimoto, Seiji; Koshiro, Tsuyoshi; Kawai, Hideaki; Oshima, Naga; Yoshida, Kohei; Urakawa, Shogo;
Tsujino, Hiroyuki; Deushi, Makoto; Tanaka, Taichu; Hosaka, Masahiro; Yoshimura, Hiromasa; Shindo,
Eiki; Mizuta, Ryo; Ishii, Masayoshi; Obata, Atsushi; Adachi, Yukimasa (2019). MRI MRI-ESM2.0 model



output prepared for CMIP6 ScenarioMIP ssp585. Version 20211003.Earth System Grid Federation.
https://doi.org/10.22033/ESGF/CMIP6.6929
Zhao, J., & Johns, W. (2014). Wind-forced interannual variability of the Atlantic Meridional
Overturning Circulation at 26.5 N. *Journal of Geophysical Research: Oceans*, *119*(4), 2403-2419.