# Peer review of "Comparing observed and modelled components of the Atlantic Meridional Overturning"

_EGUsphere, 2023_

## Author Response (AR1)

Response to Referee 1

Anonymous Referee #1, 14 Dec 2023

The manuscript is well written and easy to follow. However, I failed to understand the novelty of this study compared to that of Asbjornsen and Arthur (2023). Both use a decomposition of the AMOC to assess its decline under the SSP5-8.5 climate change scenario at 26.5°N using a similar number of CMIP6 models. Asbjorsen and Arthur (2023) do not use RAPID observations, but come to similar conclusions anyway. I do not doubt that both studies were done independently, but a clearer justification of what is new in this study and how its results and methods differ from Asbjornsen and Arthur (2023) is needed before recommending it for publication.

**We reference the Asbjørnsen and Årthun paper as confirmation that the AMOC decline and western boundary current decline in CMIP6 are robust and not just dependent on the set of specific models we used. We agree that our results and the Asbjørnsen and Årthun results on the AMOC decline in CMIP6 models over the 21st century are similar.**

**The 2 principal additions within our paper vis a vis the Asbjørnsen and Årthun paper are:**

**First, we compared RAPID components of the overturning for the period 2004-2014 with CMIP6 components for the same period lending credibility to the CMIP6 circulation while pointing out differences particularly in the deep southward flow of Lower North Atlantic Deep Water that is nearly absent in the models.**

**Secondly, we discussed the reasons for the decline in AMOC in the CMIP6 models both in terms of the trends in deep water formation and transport within the North Atlantic and also in terms of the decline in the global overturning outside the North Atlantic that appears to be due to smaller diffusive upwelling in the interior Indo-Pacific as reported by Baker et al.**

**I (Bryden) met Helene Asbjørnsen at the Royal Society meeting "Atlantic overturning: new observations and challenges" in December 2022 where we each presented posters on the Asbjørnsen and Årthun work and on the Bryden et al work. Abstracts for these posters were contained in the meeting programme. I talked to Helene at her poster and I remember suggesting that she look at variations in wind stress curl to assess changes in wind-driven circulation over the 21st century, which she subsequently did and included the results in the GRL paper. We think the Asbjørnsen and Årthun paper is excellent and our results complement their results. Our work adds a comparison between CMIP6 and RAPID components for the period 2004-2014 and discusses the implications of a decline in AMOC on both the AMOC in the North Atlantic and the extra-Atlantic MOC.**

**We have not made any specific changes to the manuscript in response to Referee 1's Comment on the relation of our work to Asbjørnsen and Årthun's paper. The 2 works are complementary as explained above.**

Possible typographical errors:

Line 77 and Table 2: Rapid instead of RAPID

Line 201: historrical instead of historical

**Thank you for pointing out some typographical errors in the submitted text. In the revised manuscript, we have corrected the references to RAPID on line 77 and Table 2 and we have corrected the spelling off historical on line 201.**

Response to Referee 2

Anonymous Referee #2, 15 Dec 2023

The AMOC strength has been observed by the RAPID array in the subtropical Atlantic since 2004. These measurements have greatly increased our understanding of AMOC dynamics and variability. They are also a critical comparator for ocean and climate models. Typically, comparisons between observations and models are of basic metrics such as the maximum and variability of the overturning streamfunction and its vertical distribution. But the observations allow far more mechanistic comparisons to the models and that is what is done here for the latest group of CMIP6 models. This paper identifies how the circulation in observations may be compared to the that in models accounting for the fact models and the real world have different zonal structures due to model resolution particularly at the Florida Straits.

The AMOC metric is a zonally integrated view of the circulation. In this paper key mechanisms are identified: Florida Straits transport, Antilles Current and several key water mass layers. These are all mechanistically interesting being relatable to both wind forcing and thermohaline forcing, particularly the wind forcing of course.

For the period 2004 to 2014 the observations and models are compared. Based on insights gained from this comparison model trends 2015-2100 are then considered and discussed.

Some important results emerge:

1. The observational community needs to refine its data products to allow better comparison with the models. Specifically, careful consideration needs to be given to how the thermocline wind-driven circulation and the western boundary current (Florida Current and Antilles Current) are separated. This is because the coarse model resolutions do not resolve the narrow Florida Straits and their western boundary currents are the sum of Florida and Antilles Currents.
2. The observations and models when carefully compared in the period 2004 to 2014 have similar mean transports overall. There are a number of caveats relating to the depth structure of the North Atlantic deep-water layers, which are considered a result of model inadequacies in representing exchanges across the Greenland-Scotland Ridge.
3. The slowing of the AMOC in the 21 century is quantified as a 30% reduction in the mean. Florida and Antilles current transport reduces by 11Sv. The wind-driven Sverdrup transport in the thermocline reduces by 4.4Sv so the wind-driven upper ocean circulation reduces by 6.6Sv. This reduction is driven by a 14% reduction of the wind-stress curl in the CMIP models. The additional necessary southward reduction in transport to achieve a mass balance across the section is 6.4Sv is in the NADW layer (in the model the reduction is mainly in upper North Atlantic Deep Water, whereas the reduction seen in observations in the early part of the 21 century occurs in the lower North Atlantic Deep Water).

This rather nice third result diagnosing the buoyancy forced reduction in AMOC is discussed with reference to Baker et al (2023). Baker show that in CMIP models this reduction is supported by reductions in the Indo-Pacific upwelling of North Atlantic Deep Water.

**Thank you for your comments reinforcing key parts of our manuscript**

While the AMOC functioning is important it is perhaps more important to additionally understand how heat transport is affected by the changing wind and buoyancy driven circulations. This is briefly mentioned in the discussion and suggests another detailed model observation comparison study on this would be valuable.

**We have referenced the Mecking and Drijfhout paper on heat transport in CMIP6 models that has now been published.**

This is a nice, short, straightforward paper that has some long-needed analysis and hopefully points a fruitful way forward for more detailed comparisons between AMOC observations and CMIP models. The attempt to separate wind and buoyancy forced changes is nice.

**Thank you for your constructive Review.**

**Minor Comments**

Lines 95-120: I think these definitions of water masses only apply to the model and not to the observations? It is not clear.

**To clarify, we have added a sentence "For the each model we have made the following choices to define Thermocline Recirculation, Intermediate Water, Upper North Atlantic Deep Water, Lower North Atlantic Deep Water and Antarctic Bottom Water."**

Line 95-102: What is the typical longitude of the eastern edge of the Antilles Current?

**Bill Johns has made nearly 7 years (2014-2021) of moored current meter measurements out to 130 km east of 78°W (out to 75.75°W). The Antilles Current has maximum current at 360 m depth at the mooring closest to the Bahamas. A Masters student, James Bourke at University of Southampton, has analysed these time series. The core of the time-averaged Antilles Current is at 360m depth within 50 km of the eastern boundary. The offshore boundary of the core Antilles Current is at WB3 (76.5°W). Bourke has estimated the time-averaged Antilles Current transport to be 5.1 Sv with a standard deviation of 7.5 Sv. We have not yet had the opportunity to publish these results.**

Line 104-105: Why was a temperature boundary chosen – is this to do with being able to compare the correct water masses between observations and models? Why 8°C which is 900m deep in the west and 800m deep in the east? The 7°C isotherm looks flatter and at ~950m deep would have more thermocline flow?

**The decision was to use potential temperature to define the boundaries between thermocline recirculation and intermediate water and between upper and lower North Atlantic Deep Water in the CMIP6 models. This choice was motivated by the indistinct upper boundary (in depth) of lower North Atlantic Deep Water in the models. We thought 8°C was a reasonable choice in the models for the boundary between thermocline recirculation and intermediate water. In observations we agree that the 7°C might be better but this is a choice for models.**

Overall a few more explanatory lines justifying the choices would be helpful.

**We have added some details on the structure of the Antilles Current and explained why we used temperature boundary in the revised version of our manuscript.**

1. Results

Table 2 is quite confusing, and it needs a much better caption to explain it and also state the units of Sv. Also note use of upper case in Table that is referenced by lower case in the text. 1. The Antilles Current transport of 5Sv is a result from Meinen et al. 2019 and should be referenced. Note also that the Meinen result for the Antilles Transport is ±4.7±7.5Sv not 5±10Sv as stated in the results; 2.

**5 Sv Antilles transport is really based on Bourke's analysis of the moored time series measurements from 2014-2021. We agree that this analysis has not yet been published but we do not wish to reference the Bourke Masters thesis since Johns deserves much credit for developing the time series. So we use a value of Antilles transport of 5 Sv, and reference the Meinen work as we round up their transport from 4.7 to 5 Sv.**

Western Boundary Current (FS+AC) does equal 31.3+5=36.3Sv. But Thermocline Recirculation+AC -18.6+5=-13.6Sv does not equal -23.6Sv as in paper. The number you want in the table is the -23.6Sv but you need to read the results section to carefully understand your argument for separating the thermocline and Antilles transports. It needs an explanatory note in the caption or you need to write Recirculation-AC; The line Western Boundary Current+Ekman+Model TR should be written as AMOC= Western Boundary Current+Ekman+Model TR so it is obvious it can be compared directly to AMOC=FS+Ekman+IW+TR for the AMOC northward upper limb. The Deep Water part of the table is much simpler especially as the AMOC deep limb appears on the same line!

**The Referee is correct about the transport values. We have changed the labels for transport values in Table 2 to reflect Referee's statements and to clarify the argument.**

A recent paper discusses the Gulf Stream weakening which the authors might reference (Piecuch, C. G. & Beal, L. M. Robust Weakening of the Gulf Stream During the Past Four Decades Observed in the Florida Straits. *Geophysical Research Letters* **50** (2023). https://doi.org:10.1029/2023gl105170)

**We now reference the Piecuch and Beal paper and we have also added references to McCarthy and Caesar, and Ditlevsen and Ditlevsen.**

Response to Referee 3

Review of Bryden et al. "Comparing observed and modelled components of the AMOC at 26ºN"

This paper presents a comparison of CMIP6 model output vs RAPID observations, with transports decomposed similar to RAPID. I found the paper easy to follow, had some

nice results. It's quite a short paper for OS but I appreciate it came about in an unusual way with the MSc student not pursueing it.

**Thank you**

The Figure 2 is very informative, especially l182–183: this is a very nice finding. Very striking WBC decline and the decomposition of this into wind-driven components is very interesting (l212). I think one more figure should be added: a figure that simplifies Fig 2 to show MOC decline broken into wind-driven and non-wind components.

**With respect to adding a new figure with the circulation separated into wind and non-wind components, we decided not to add such a figure. The decline in the directly wind-driven MOC is exactly reflected in the 17% decline of the Thermoclne Recirculation (TR) and the decline in the non-wind MOC is reflected in the 34% decline in uNADW transport. Both are already in Figure 2. So we have added a statement of these component declines in the text and reference these components in Figure 2.**

I confess I haven't read Asbjornsen and Arthun but their abstracts sounds very similar to your paper. Can you clarify the differences and unique insights from this study?

**We have addressed the similarities and differences between our study and the Asbjørnsen and Årthun paper in our response to Reviewer 1 as follows**

**We reference the Asbjørnsen and Årthun paper as confirmation that the AMOC decline and western boundary current decline in CMIP6 are robust and not just dependent on the set of specific models we used. We agree that our results and the Asbjørnsen and Årthun results on the AMOC decline in CMIP6 models over the 21st century are similar.**

**The 2 principal additions within our paper vis a vis the Asbjørnsen and Årthun paper are:**

**First, we compared RAPID components of the overturning for the period 2004-2014 with CMIP6 components for the same period lending credibility to the CMIP6 circulation while pointing out differences particularly in the deep southward flow of Lower North Atlantic Deep Water that is nearly absent in the models.**

**Secondly, we discussed the reasons for the decline in AMOC in the CMIP6 models both in terms of the trends in deep water formation and transport within the North Atlantic and also in terms of the decline in the global overturning outside the North Atlantic that appears to be due to smaller diffusive upwelling in the interior Indo-Pacific as reported by Baker et al**

Overall, I think it's an interesting study that can be published following minor corrections.

**Thank you**

Minor comments:

l36: add a line on importance of the global overturning circulation

**We have added a (short) sentence summarising the importance of the global overturning circulation. It is a complex topic that would require extensive discussion to cover it adequately. We could provide several pages of text on the topic but such discussion would divert emphasis from the focus of this manuscript.**

l48: the latest RAPID data are available to 2022 at time of writing. Please explain why your analyses ends in 2018 and any issues that might arise because of this shortening.

**RAPID data has a time delay due to the time offset between eastern and western expeditions to recover moorings. When we started this study, RAPID data was only archived up to 2018.**

l52. If you're not going to state the mean transport in the climate models, then you need to report the decline differently i.e. not a percentage. Weijer also reports the decline as 1 Sv/decade (I think).

**The decline is stated in Table 3 both as a change in transport (in Sv) and as a percentage change. We leave it to the reader to infer the decline in Sv/decade.**

l64. I think you can bring in findings from a few more papers than just Weijer here:

Robson, Jon, et al. "The role of anthropogenic aerosol forcing in the 1850–1985 strengthening of the AMOC in CMIP6 historical simulations." Journal of Climate 35.20 (2022): 6843-6863.

McCarthy, Gerard D., and Levke Caesar. "Can we trust projections of AMOC weakening based on climate models that cannot reproduce the past?." Philosophical Transactions of the Royal Society A 381.2262 (2023): 20220193.

**We have added references to Piecuch and Beal, McCarthy and Caesar, and Ditlevsen and Ditlevsen. We consider the Robson et al. paper to be less relevant here because it is about historical changes in the AMOC and not about future changes that are the focus of our work.**

l83. Why monthly?

**RAPID time series were averaged to 30-day values to match model time series. It is common practice within the RAPID community to use time series of 30-day average values.**

l87. How was the one ensemble member chosen? Was it the first one?

**The choice of ensemble member is indicated in Table 1 and the prefereed ensemble member was realisation 1, initialisation 1, physics 1 and forcing 1, indicated by r1i1p1f1. For some models forcing 1 was not available so a different**

**ensemble member was chosen making sure that the forcing version (v6.2.0) is the same. These choices are now described in the Caption for Table 1.**

l91. Can you show the range of the net transport through the section? There is surely quite a range around -1Sv.

**The range of net transport is from -0.5 Sv to -1.75 as shown in Beunk, 2022, Appendix A. In our opinion, it is not a large range about 1 Sv.**

l104–118. Why use isotherms rather than depth as RAPID does?

**The decision was to use potential temperature to define the boundaries between thermocline recirculation and intermediate water and between upper and lower North Atlantic Deep Water. This choice was motivated by the indistinct upper boundary (in depth) of lower North Atlantic Deep Water in the models.**

l120. Is Ekman recalculated in the models from their winds? Or taken as very near surface ageostrophic transport?

**Ekman is calculated in the models from the model wind stress.**

Figure 1: I'm surprised at the small spread of the climate models mean AMOC. Was there any selection of the models based on mean AMOC strength?

**We are also surprised by the small spread. Models were not selected on the basis of mean AMOC. As explained by Beunk (2022), "the choice of these models is motivated by the fact that historical and SSP85 date is available for all used variables" including meridional velocity, zonal wind stress, salinity and temperature. In addition only models that use horizontal depth values are included. Choice of models is now described in the caption to Table 1**

**Thank you for your careful review of our manuscript.**